# Computational Fluid Dynamics (CFD) Simulations and Experimental Measurements in an Inductively-Coupled Plasma Generator Operating at Atmospheric Pressure: Performance Analysis and Parametric Study

**Sangeeta B. Punjabi** [1,2] , **Dilip N. Barve** [3] , **Narendra K. Joshi** [4], **Asoka K. Das** [3],
**Dushyant C. Kothari** [2] , **Arijit A. Ganguli** [5,6] , **Sunil N. Sahasrabhude** [3,*] **and**
**Jyeshtharaj B. Joshi** [5,7,*]

[1] Electrical Engineering Department, V.J.T.I, Matunga, Mumbai 400019, India; p.sangeeta@gmail.com
[2] Department of Physics, University of Mumbai, Kalina, Santacruz (E) 400098, India; kothari@gmail.com
[3] Laser and Plasma Technology Division, BARC, Mumbai 400085, India; dnbarve@gmail.com (D.N.B.);
   asokadas@gmail.com (A.K.D.)
[4] Department of Nuclear Science and Technology, Mody University of Science and Technology,
   Lakshamangarh (Sikar) 332311, Rajasthan, India; nkjoshi50@gmail.com
[5] Department of Chemical Engineering, Institute of Chemical Technology, Matunga, Mumbai 400019, India;
   ganguliarijit@gmail.com
[6] Department of Chemical Engineering, School of Engineering and Applied Sciences, Ahmedabad University,
   Ahmedabad 380009, India
[7] Homi Bhabha National Institute, Anushaktinagar, Mumbai 400094, India
[*] Correspondence: snsahasrabudhe@gmail.com (S.N.S.); jbjoshi@gmail.com (J.B.J.)

**Abstract:** In this article, electrical characteristics of a high-power inductively-coupled plasma (ICP) torch operating at 3 MHz are determined by direct measurement of radio-frequency (RF) current and voltage together with energy balance in the system. The variation of impedance with two parameters, namely the input power and the sheath gas flow rate for a 50 kW ICP is studied. The ICP torch system is operated at near atmospheric pressure with argon as plasma gas. It is observed that the plasma resistance increases with an increase in the RF-power. Further, the torch inductance decreases with an increase in the RF-power. In addition, plasma resistance and torch inductance decrease with an increase in the sheath gas flow rate. The oscillator efficiency of the ICP system ranges from 40% to 80% with the variation of the Direct current (DC) powers. ICP has also been numerically simulated using Computational Fluid Dynamics (CFD) to predict the impedance profile. A good agreement was found between the CFD predictions and the impedance experimental data published in the literature.

**Keywords:** inductively-coupled plasma; impedance; plasma; energy

## 1. Introduction

Inductively-coupled plasma (ICP) discharge has been widely used for materials processing applications such as etching, the synthesis of ultrafine powders of metals, and powder spherodization. ICP is also used in spectrochemical analysis, nano-particle synthesis, plasma processing and deposition of thin films [1–4]. One of the most important issues in the investigation for ICP discharge is to find the process parameters that affect the characteristics of plasma [5–7]. Further, to design a radio-frequency (RF) generator, knowledge of plasma impedance and its variation under different operating conditions is of great importance. Plasma impedance can be calculated by solving Maxwell equations with the generator voltage being used as the input source for the electromagnetic field [8]. The fundamental

governing equations in the inductor and plasma region of the ICP are solved by the finite difference or finite element methods. The solution of these equations is challenging since the number of unknown variables are more than the number of unknown equations and requires suitable approximations. Moreover, the strong coupling between momentum, energy, and Maxwell equations inside the plasma region makes the problem computationally intensive for industrial applications.

To address the above problem, various models have been proposed by various authors [8–11]. A hybrid boundary element-finite difference method together with 2D simulation was used by Fouladgar and Chentouf [9] to predict the resistance and inductance as a function of temperature in radio-frequency inductively-coupled plasma (RF-ICP) operating in the range of 3–5 MHz, using argon as plasma gas. The authors have suggested that to reduce non-linearity complications, the bloc Diagonal Gauss–Seidel method should be used. The model was later tested by Chentouf et al. [8] at a frequency of 3–5 MHz. The authors have reported that an exponential distribution of current density in the conductor and parabolic velocity profile in the axial direction gave good predictions. The models, however, do not go through the computation of full flow and temperature fields in the discharge and hence do not include the inherent changes in the electromagnetic properties of plasma which may result in changes in the electromagnetic properties of the generator circuit. Kim et al. [10] devised an iterative method by which the ICP was simulated as part of the RF-circuit. They assumed that the steady-state signal of a feedback oscillator circuit is independent of the transient conditions. By this method, they could find the overall efficiency of power transmission to plasma, plasma impedance, and velocity and temperature fields inside the plasma torch. The model, however, relies on a relatively simple laminar flow model which does not permit the computation of flow and temperature fields in the discharge under realistic gas flow and high-power conditions. Further, the authors do not report any information on the flow and temperature fields in the discharge associated with the change of the electrical power supply setting. An integrated model was proposed by Merkhouf and Boulos [11] to predict the overall electrical characteristic of RF-ICP using a 2D turbulence model. The authors claimed that the model was comprehensive and the overall characteristics of the system were coupled to the temperature, flow, and gas composition fields. This model was later experimentally validated by Merkhouf and Boulos [12], and a good agreement with the experimental data was observed. The proposed model was, however, tested with a constant sheath gas flow rate which is an important process parameter. A summary of the previous work for calculating impedance by various authors, their findings, limitations, and assumptions have been presented in Table 1. Recently, several ICP modeling studies on three-dimensional (3D) time-dependent RF-ICP have been reported [13–15]. The works mainly focus on complex coherent vortex structures and dynamic behaviors produced near the coil region in the ICP. These are unsteady in nature and require Large Eddy Simulation (LES) turbulence models for understanding. These are out of the scope of the present work.

The literature review shows that continuous progress has been made in numerical modeling for the prediction of overall characteristics of the RF-generator system with flow and temperature fields. However, models [8–11] have not yet been tested for the variation of the sheath gas flow rate which is an important process parameter. Further, a wide range of powers that are necessary to check the oscillator efficiency needs close attention. The objective of the present study is to determine how the plasma resistance and the torch inductance vary with RF power and sheath gas flow rates. Experimental investigations to understand the variation of efficiency as a function of Direct Current (DC) power for various gas flow rates in an ICP torch have been carried out [12]. In the present investigation, effort has been made to present a CFD simulation model which can be used to predict the parametric sensitivity discussed above. The CFD predictions have been compared with the available data in the published literature, as well as the experimental data of the present work, over a wide range of the sheath gas flow rate and the power input.

**Table 1.** Literature review.

| Authors | Geometry | Method | Assumptions † | Findings ‡ | Conclusions # | Limitations ## |
|---|---|---|---|---|---|---|
| Fouladgar and Chentouf [8] | 2D axi-symmetric geometry with frequency variable from 3 to 5 MHz. Energy and electromagnetic equations are solved | Hybrid Finite Element-Boundary Element (FE-BE) | 1, 2 | 1,2,3 | 1,2 | 1, 2, 3 |
| Chentouf et al. [9] | 2D axi-symmetric geometry with frequency variable from 3 to 5 MHz. Energy and electromagnetic equations are solved | Boundary elements-Finite Difference | 1, 2, 3 | 1, 4, 5, 6 | 3, 4 | 1, 4 |
| Kim et al. [10] | 2D model of induction plasma generator in which ICP is considered part of RF network. Simulations were carried out using two frequencies (2.39 and 4.95 MHz) | Non-linear steady state approach | 1, 4, 5, 6, 7, 8 | 7, 8, 9 | 5, 7 | 3, 5 |
| Merkhouf and Boulos [12] | 2D model is solved using k-ε turbulent model with non-linear analytical model of the generator circuit | Integrated model | 4, 5, 8 | 1, 10, 11, 12, 13, 14, 15 | 8 | 3, 6 |
| Merkhouf and Boulos, [11] | 2D model is solved using k-ε turbulent model. This work was to validate the earlier published simulated results | Integrated model | 4, 5, 8 | 16, 17, 18 | 9 | 7 |

† **Assumptions: 1.** The flow field is assumed to be laminar in the z direction. **2.** As the velocity is zero at the wall of the quartz tube, a parabolic variation of velocity is considered. **3.** An exponential decay for the current distribution inside the inductor with skin depth has been assumed. **4.** Local thermodynamic equilibrium. **5.** The plasma is optically thin. **6.** Steady state. **7.** Negligible displacement currents and viscous dissipation. **8.** Axisymmetric. ‡ **Findings: 1.** Radial temperature profile was estimated. **2.** Coil currents at each turn are found. **3.** Resistance and inductance of the system as a function of temperature. **4.** Axial temperature distribution and radial power distribution has been found. **5.** Impedance with plasma and without plasma has been given. **6.** The different powers and losses in different parts of the system have been estimated. **7.** The steady state signals from the triode's plate and the grid were obtained. **8.** The efficiencies of RF generator as a function of plate bias voltage is predicted. **9.** Impedance profiles (plasma resistance and reactance) as a function of plasma power and frequency are obtained and explained. **10.** Predicted coil current versus the plate bias voltage. **11.** Temperature contours for 6 and 9 kV plate voltage. **12.** Variation of equivalent inductance and resistance with plate voltage and its explanation. **13.** Profile of anode loss and oscillator efficiency with plate voltage has been obtained. **14.** Variation of torch efficiency and overall coupling efficiency as a function of plate power is predicted and explained. **15.** Effect of plasma gas flow rate on temperature contours and the ratio of turbulent to molecular viscosities in the discharge has been obtained. **16.** The energy distribution for 6.5, 7.0, and 7.5 kV dc plate voltage is given. There is reasonably good agreement between model and experimental measurement. **17.** Dc plate current and RF coil current as a function of plate voltage have been given. The maximum difference between the model and the experimental measurement is found at higher plate voltage (7.5 kV). **18.** Impedance drops with the increase in plate voltage. # **Conclusions: 1.** The current distribution is highly non-homogeneous. **2.** The resistance of the torch increases and the reactance decreases with the increase in temperature. **3.** Temperature and current distribution in induction plasma has been studied using a Boundary-Elements Finite-Difference method. **4.** The calculation time for this algorithm is smaller than the direct method. **5.** A mathematical model was developed to predict the overall behavior of the RF plasma generator. **6.** A 2D vector potential model was modified to compute impedance of plasma torch. **7.** By altering the computation of the transient response and steady state flow, the frequency, steady state output signal of triode, and plasma power were obtained. **8.** The integrated model proposed for representation of flow, temperature, and electromagnetic parameters of induction plasma torch and RF power system. **9.** The predictions of an integrated model and the experimental (electrical and calorimetric) results show a good agreement. ## **Limitations: 1.** Variation in frequency from 3 to 5 MHz is considered. **2.** Variation of resistance and reactance of the torch with temperature is given. **3.** No experimental validation of impedance (resistance and reactance) of the torch has been done. **4.** Impedance of the torch has been found for two extreme (without plasma and with plasma) conditions only. **5.** Transient dynamics of the plasma have been neglected. **6.** From the formulation of equivalent plasma resistance and inductance, it is not clear whether it is for plasma or torch. **7.** No variation of plasma resistance or inductance has been mentioned.

## 2. Experimental Set-Up

The circuit diagram of the 50 kW RF oscillator is shown in Figure 1A. The high-power RF oscillator is a Colpitts type. It uses a triode tube (1) (BW 1608J2F) as an amplifier. It is powered by a DC supply (2) with a 120-kW rating. The supply can generate a DC voltage of up to 12 kV and a DC current of up to 10 A. This supply is isolated from the Alternating Current (AC) voltage generated by the oscillator with the help of two inductors (3), one connected to the plate of the triode and one connected to the cathode of the triode. In addition to this, a capacitor $C_f$ (4) is used to protect the DC supply from AC voltages generated at the plate of the triode. The cathode is indirectly heated by a filament. Filament heater supply is derived from a step-down transformer. Filament voltage and current are measured using a voltmeter $V_f$ (5) and a current transformer $C_T$ (6), respectively. A small capacitor $C_{fm}$ (7) bypasses any high-frequency voltage developed across the filament ends.

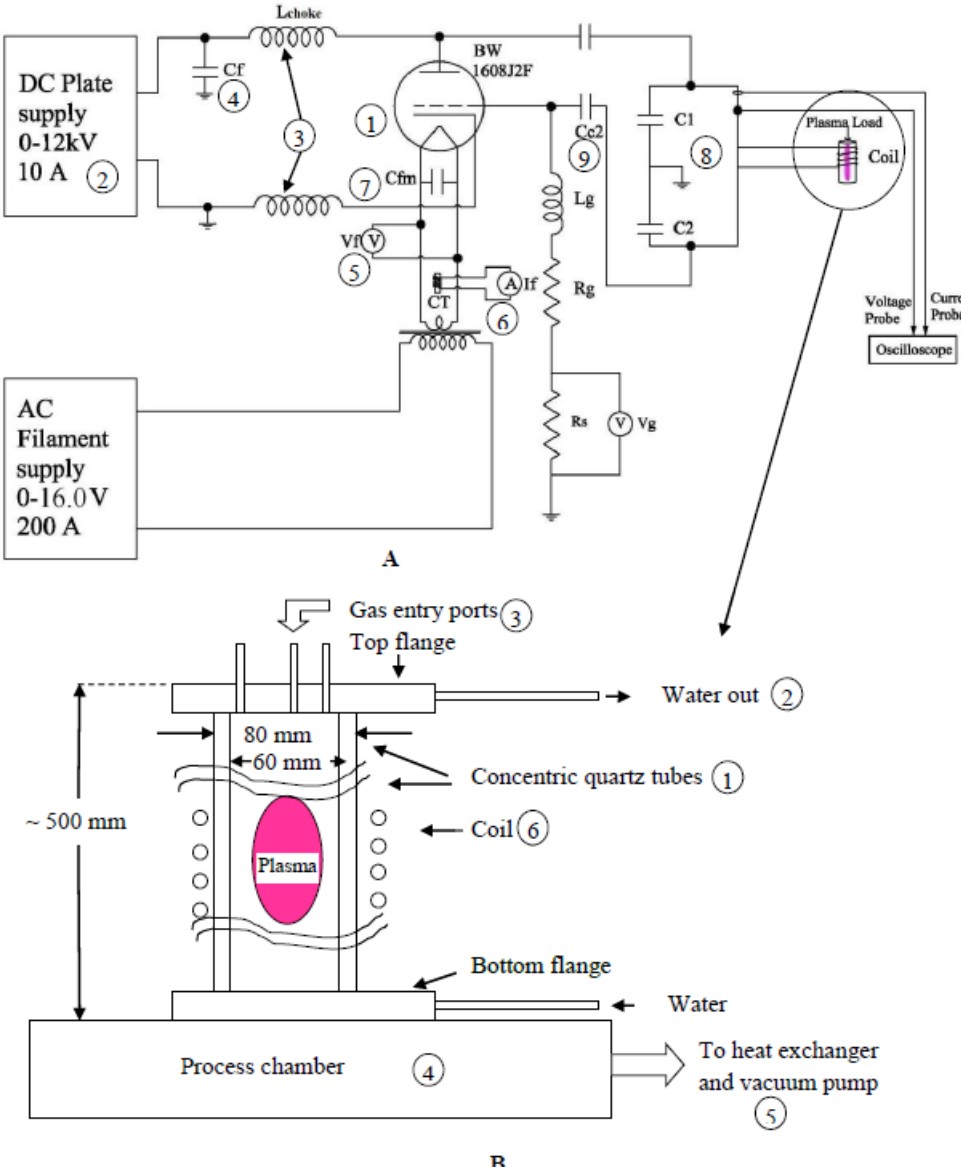

**Figure 1.** Schematic diagram of 50 kW RF Oscillator circuit. (**A**) 1. Triode tube; 2. DC Supply; 3. Two inductors; 4. Capacitor to protect DC supply from AC voltage; 5. Voltmeter; 6. Current Transformer; 7. Capacitor to bypass high frequency voltage; 8. Voltage capacitors; and 9. Capacitor. (**B**) Schematic of the inductively-coupled plasma (ICP) torch. 1. Concentric Quartz tubes; 2. Cooling water; 3. Gas ports; 4. Process chamber; 5. Heat exchanger and vacuum pump; 6. Induction coil.

The resonant tank circuit consists of a coil (with inductance $L_C$) and high-voltage capacitors C1 and C2 (8). Specifications of the tank circuit and other components are given in Table 2. A part of the high-frequency voltage across the tank circuit is derived from the potential divider formed by the capacitors C1 and C2 and is fed back to the grid through capacitor C2 (9). The grid is connected to the ground through a grid leak resistance $R_g$ and grid choke $L_g$. $R_g$ develops the required grid bias. The tank coil and the plate of the valve are water-cooled.

**Table 2.** Specifications of the tank circuit components.

| Components | Values |
|---|---|
| Inductance of Coil ($L_C$) | 1.59 μH |
| Resistance of coil | 0.03 ohm |
| Number of turns of coil | 4 |
| Coil Outer diameter | 110 mm |
| Coil Inner diameter | 90 mm |
| C1 | 11.86 nF |
| C2 | 1.42 nF |
| Grid leak resistance $R_g$ | 330 Ω |

An ICP torch is made up of two concentric quartz tubes (1), as shown in Figure 1B. Cooling water (2) flows between the quartz tubes. The flow rate of the water is minimal and streamlined, hence it cannot generate its own inductive field. For the water to boil, the temperature should be around 373 K, but in our experiments, the temperature of quartz tube facing water is less than about 330 K. The water flow rate is adjusted in such a manner that its temperature does not exceed 330 K, hence the water flowing through concentric pipes does not boil. The diameter of the inner tube (ID) is 60 mm and the diameter of the outer tube (OD) is 80 mm. The torch is vertically mounted on a process chamber (3) and gases are introduced from the top through gas ports (3). A rotary vacuum pump (5) is attached to the process chamber through a heat exchanger (5). The pressure in the chamber is adjusted by control valves.

Initially, the torch is evacuated to a low pressure (~50 mbar) and a DC voltage of about 2 to 3 kV is applied to the triode. At this point, a glow discharge starts due to oscillating current developed in the coil (6). After adjusting the DC current, voltage, and the gas flow, arc discharge strikes and highly luminous plasma are generated. After this, the pressure is slowly raised to 1000 mbar. Gas flows and DC power fed to the ICP torch are adjusted as per the requirement of the experiment.

The resonating frequency is given by [16]:

$$f = \frac{1}{2\pi \sqrt[2]{L_T C}} \left( 1 - \frac{R_T^2 C}{L_T} \right) where\ C = \frac{C1 C2}{C1 + C2}$$
$$L_T = L_C - L_P,\ R_T = R_C + R_P \tag{1}$$

where $L_T$ is inductance of the torch, $L_C$ is the coil inductance, $L_P$ is the inductance of plasma, $R_T$ is the resistance of the torch, $R_C$ is the resistance of the coil, and $R_P$ is the reflected resistance of plasma as seen by the coil. $L_P$ neutralizes a portion of primary inductance $L_C$, thereby reducing the equivalent inductance that is observed between the terminals of the primary coil [17].

## 3. Experimental Measurements

### 3.1. Calorimetric Measurements

Cooling loops to different parts of the system are illustrated in Figure 1. A pump (1) draws water from the storage tank (2) and passes it through a heat exchanger (3). Water thus cooled is directed back to the tank.

Another pump (4) draws water from the storage tank (2) and directs it to the inlet header (5). Three water lines (6) are taken from the inlet header. One of the lines cool the plate of the triode

(oscillator) (7), other cools the inductor coil (8) (which excites the plasma) and third the ICP torch (9). These three water lines again join at the outlet header (10) and the water is directed back to the storage tank. A mercury thermometer (11) and a calibrated water rotameter (12) are used in each cooling water line to measure the temperature rise ($\Delta T$) and mass flow rate ($\dot{m}$) of cooling water through each section of the cooling system. Inlet water temperature is measured by the thermometer $T_{in}$ (13). The power loss ($P_i$) in cooling water for each element $i$ (plate of the triode torch, induction coil and torch wall) is calculated as follows:

$$P_i = \dot{m}_i C_p (T_o - T_{in}) \tag{2}$$

Power loss in the grid circuit is given by:

$$P_g = I_g^2 R_g \tag{3}$$

### 3.2. Electrical Measurements

The electrical measurements, namely the RF current and voltage measurements have been carried out with a Rogowski coil similar to that which was used in the study of Merkhouf and Boulos [11]. The RF power measurement is described below:

All the DC operating parameters such as, DC plate voltage ($V_{dc}$), plate current, and grid current ($I_{dc}$) are monitored on the control panel. The DC power supplied to the oscillator tube is given by:

$$P_{dc} = V_{dc} I_{dc} \tag{4}$$

Measurement of the RF voltage was done using a Tektronix high-voltage probe (P6015A) connected across C1, as shown in Figure 2. Knowing the multiplication factor of the probe, $V_c$ across the coil was calculated. The Root Mean Square (RMS) RF current across the coil ($I_c$) was measured using a Rogowski coil (Power Electronic Measurements Ltd., Nottingham, UK) having sensitivity of 4.16 mV/Amp. Values of $I_c$, $V_c$, and $f$ are measured using a Digital Storage Oscilloscope (DSO). The RF power produced is given by:

$$P_{RF} = V_c I_c \cos\theta \tag{5}$$

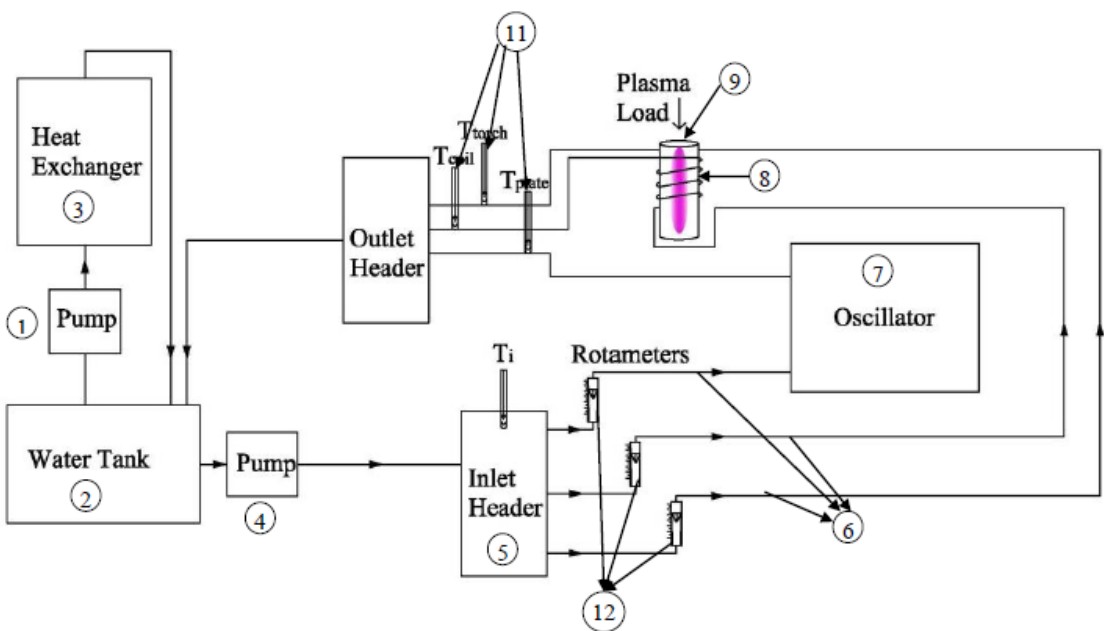

**Figure 2.** Schematic of water cooling arrangement for calorimetric measurements. 1. Pump, 2. Storage tank, 3. Heat exchanger, 4. Pump, 5. Inlet header, 6. Water lines, 7. Oscillator, 8. Inductor coil, 9. ICP torch, 10. Outlet header, 11. Mercury Thermometer, 12. Rotameter.

Here, we assume that almost all of this power ($P_{RF}$) is deposited in plasma and other losses are negligible. Typically, $V_c$ is approximately a few kV and $I_c$ is over 100 A. Table 3 shows $V_c$ and $I_c$ data recorded for a 3 lpm plasma gas flow rate and different sheath gas flow rates. To get a reasonable value of RF power from Equation (5), one should estimate the range of $\theta$. We can do this by taking two extreme cases:

**Table 3.** Electrical measurements on an inductively-coupled plasma (ICP) torch for plasma gas flow rate of 3 lpm.

| Sheath Gas (lpm) | DC Power (kW) | $V_c$ (kV) | $I_c$ (A) | RF Power (kW) | $\theta$ in Deg. (Estimated) |
|---|---|---|---|---|---|
| 10 | 21.25 | 3.74 | 112.02 | 15.57 | 87.87 |
|  | 15.0 | 3.11 | 93.75 | 10.39 | 87.96 |
|  | 9.75 | 2.67 | 79.23 | 6.37 | 88.28 |
|  | 5.5 | 2.21 | 63.77 | 3.04 | 88.76 |
|  | 3.75 | 2.08 | 59.20 | 1.50 | 89.30 |
| 15 | 21.25 | 3.65 | 106.92 | 16.39 | 87.59 |
|  | 15.0 | 3.10 | 91.29 | 11.52 | 87.67 |
|  | 9.75 | 2.60 | 76.37 | 7.14 | 87.94 |
|  | 5.0 | 1.00 | 58.26 | 3.22 | 88.41 |
|  | 3.75 | 1.97 | 55.81 | 1.97 | 88.97 |
| 21 | 21.25 | 3.64 | 108.43 | 15.90 | 87.69 |
|  | 12.25 | 2.79 | 83.62 | 8.80 | 87.84 |
|  | 7.5 | 2.13 | 62.88 | 4.80 | 87.95 |
|  | 5.0 | 2.01 | 57.88 | 2.55 | 88.74 |
|  | 3.75 | 1.86 | 53.34 | 1.53 | 89.11 |
| 25 | 21.25 | 3.62 | 115.74 | 16.53 | 87.74 |
|  | 15.0 | 3.07 | 97.95 | 11.33 | 87.84 |
|  | 9.0 | 2.49 | 78.48 | 5.76 | 88.31 |
|  | 5.0 | 2.08 | 63.99 | 2.55 | 88.90 |
| 30 | 22.5 | 4.05 | 125.19 | 16.88 | 88.09 |
|  | 16.0 | 3.39 | 105.43 | 11.53 | 88.15 |
|  | 9.0 | 2.51 | 77.52 | 5.98 | 88.24 |
|  | 5.0 | 1.93 | 58.19 | 2.54 | 88.70 |
|  | 3.75 | 1.88 | 56.05 | 1.36 | 89.26 |

Case (A): If, the RF power produced $P_{RF} = 0$, then $\theta = 90°$ and Case (B): If, RF power produced $P_{RF}$ = DC power supplied to the oscillator, then values of $V_c$ and $I_c$ are taken from Table 3. Hence, we take data for $V_c$ and $I_c$ from Table 3 for a DC power of 22.5 kW. Substituting the values of $V_c$, $I_c$, and $P_{RF}$ into Equation (5) we obtain $\theta = 88.09°$. An effort was taken to measure $\theta$ directly using a DSO. However, either due to the presence of harmonics or the dynamic nature of plasma load, correct measurement of $\theta$ was not possible by direct DSO measurement in the present system. For more accuracy in the measurement of RF power, Case (C) was considered. In Case (C), the power losses in the grid circuit and in the plate circuit of the triode have been considered and subtracted from the applied DC power to give better values of the RF power. Thus, in comparison to Case (B), the RF power obtained by Case (C) is accurate in the calculation of $\theta$. Further, the ratio of actual RF power measured to the DC power fed via DSO will give us the oscillator efficiency. Thus, we define the oscillator efficiency as follows:

$$Oscillator\ efficiency\ (\%)\ =\ \frac{P_{dc} - \left( P_g + P_{plate} \right)}{P_{dc}} \times 100 \tag{6}$$

### 3.3. Error Analysis

The experimental measurements were made by the instruments having finite least counts which cause errors in measurements. Using a standard error analysis technique, the uncertainty in various measurements was estimated. The source of error and their least count are as follows:

1.  DC power supply: Deviations in the voltage and current panels are given below:

    (a)   Voltage panel: 0.25 kV
    (b)   Current panel: 0.25 A

    The combined error was calculated by the following formula:

    $$\text{Power} = V \times I = (V \pm \Delta I) \times (I \pm \Delta I)$$

    The voltage V = 5 KV for the present case and current I = 4 A

    From the deviations given above, the errors in the voltage and current, respectively, are 5% and 6.25%. The large errors are due to the magnitudes of power involved. For example, if we consider 20 kW power, 5 kV $\times$ 4 A = 20 kW has error due to least count of meters showing the readings. When we use the standard formula for error that is $(V + \Delta V) \times (I + \Delta I)$ where $\Delta V$ and $\Delta I$ are the least count of the display meters, we get 20 + 2.3125 which turns out to be resulting in an error of about 11.5%.

2.  Water flow meter:

    An error of 0.5 lpm was noted. The flow rate was 5 lpm. Hence, the percentage error was 10%.

3.  Water temperature:

    An error of 0.5 °C was noted. The temperature difference ($\Delta T$) for one cooling circuit was 4 °C; hence, the error in $\Delta T$ is around 10%.

4.  Grid Power:

    Rid voltage: 0.1 V. Since the power loss in the grid is very small compared to the plate loss and other losses, the error in its measurement does not contribute much to the error in the calculation of dissipated power and thus in the calculation of oscillator efficiency.

    It was observed that the percentage error in the efficiencies and RF power is high at low DC power and low at high DC power. Rounding off and truncation errors are very small compared to errors due to finite least counts and, therefore, they are ignored. The overall experimental errors accounted for 20–25%.

## 4. Mathematical Modeling

### 4.1. Governing Equations, Assumptions, and Boundary Conditions

The governing equations (continuity, momentum, energy, and vector potential equations) are solved in with Ansys Fluent 14. A user-defined scalar (UDS) is applied for solving the vector potential equations. All the governing equations and the boundary conditions for solving the governing equations have been described in Table 4.

**Table 4.** Governing equations in cylindrical co-ordinates for axi-symmetric flow.

| Property | Equations |
|---|---|
| Continuity | $\frac{1}{r}\frac{\partial}{\partial r}\langle r\vartheta_r\rangle + \frac{\partial}{\partial z}\langle\vartheta_z\rangle = 0$ |
| Axial velocity Momentum | $\langle\vartheta_z\rangle\frac{\partial\langle\vartheta_z\rangle}{\partial z} + \frac{1}{r}\langle v_r\rangle\frac{\partial\langle r\vartheta_z\rangle}{\partial r} = -\frac{1}{\rho}\frac{\partial\langle p\rangle}{\partial z} + (\nu+\nu_t)\left[\frac{1}{r}\frac{\partial}{\partial r}\left(\frac{\partial\langle r\vartheta_r\rangle}{\partial r}\right) + \frac{\partial}{\partial z}\left(\frac{\partial\langle\vartheta_z\rangle}{\partial z}\right)\right] + F_z$ |
| Radial velocity Momentum | $\frac{1}{r}\langle u_r\rangle\frac{\partial\langle r\vartheta_r\rangle}{\partial r} + \langle u_z\rangle\frac{\partial\langle\vartheta_r\rangle}{\partial z} = -\frac{1}{\rho}\frac{\partial\langle p\rangle}{\partial z} + (\nu+\nu_t)\left[\frac{1}{r}\frac{\partial}{\partial r}\left(\frac{\partial\langle r\vartheta_r\rangle}{\partial r}\right) + \frac{\partial}{\partial z}\left(\frac{\partial\langle\vartheta_z\rangle}{\partial z}\right) - \frac{\vartheta_r}{r^2}\right] + F_r$ <br> where, $F_r = \frac{1}{2}\mu_0\sigma Real\left[E_\theta H_z^\times\right]$; $E_\theta = -i\omega\,A_\theta$; $\mu_0\,H_z = \frac{1}{r}\frac{\partial}{\partial r}(rA_\theta)$ and <br> $\mu_0\,H_r = -\frac{\partial}{\partial z}(A_\theta)$ |
| Energy | $\frac{1}{r}\frac{\partial\langle\vartheta_r H\rangle}{\partial r} + \frac{\partial\langle\vartheta_z H\rangle}{\partial z} = \frac{1}{r}\frac{\partial}{\partial r}\left[r\alpha_{\text{eff}}\frac{\partial\langle H\rangle}{\partial r}\right] + \frac{\partial}{\partial z}\left[\alpha_{\text{eff}}\frac{\partial\langle H\rangle}{\partial z}\right] + U_P + U_R$ <br> $U_P = \frac{1}{2}\sigma\left[E_\theta E_\theta^\times\right]$ $U_R = 5600(T-9500) + 181(T-9500)^2$ |
| Turbulent Kinetic Energy | $\frac{1}{r}\langle\vartheta_r\rangle\frac{\partial(r\kappa)}{\partial r} + \langle\vartheta_z\rangle\frac{\partial\kappa}{\partial z} = \left(\nu + \frac{\nu_t}{\sigma_\kappa}\right)\left[\frac{1}{r}\frac{\partial}{\partial r}\left(\frac{\partial(r\kappa)}{\partial r}\right) + \frac{\partial}{\partial z}\left(\frac{\partial\kappa}{\partial z}\right)\right] + G_\kappa - \rho\varepsilon$ <br> Turbulent kinetic energy $G_\kappa = \nu_t\left|\overline{S}\right|^2$ where $\left|\overline{S}\right| = \sqrt{2\overline{S_{ij}S_{ij}}}$ and $\left|\overline{S}\right| = \frac{1}{2}\left(\frac{\partial\vartheta_z}{\partial r} + \frac{\partial\vartheta_r}{\partial z}\right)$ |
| Energy Dissipation Rate equation | $\frac{1}{r}\langle v_r\rangle\frac{\partial(r\varepsilon)}{\partial r} + \langle v_r\rangle\frac{\partial\varepsilon}{\partial r} = \left(\nu + \frac{\nu_t}{\sigma_k}\right)\left[\frac{1}{r}\frac{\partial}{\partial r}\left(\frac{\partial(r\varepsilon)}{\partial r}\right) + \frac{\partial}{\partial z}\left(\frac{\partial\varepsilon}{\partial z}\right)\right] + C_{\varepsilon_1}\frac{\varepsilon G_k}{k} - C_{\varepsilon_2}\frac{\rho\varepsilon^2}{k}$ <br> $C_{\varepsilon_1} = 1.44$, $C_{\varepsilon_2} = 1.92$ |
| Vector Potential Equation | $\frac{\partial^2 A_R}{\partial z^2} + \frac{1}{r}\frac{\partial}{\partial r}\left(r\frac{\partial A_R}{\partial r}\right) - \frac{A_R}{r^2} + \mu_0\omega\sigma\,A_I = 0$ <br> $\frac{\partial^2 A_I}{\partial z^2} + \frac{1}{r}\frac{\partial}{\partial r}\left(r\frac{\partial A_I}{\partial r}\right) - \frac{A_I}{r^2} - \mu_0\omega\sigma\,A_R = 0$ <br> $A_\theta = A_R + iA_I$ |

The assumptions made are as follows:

1. Flow field is affected by local plasma temperature changes.
2. System is axially symmetric.
3. Steady-state, incompressible, turbulent flow.
4. Negligible viscous dissipation.
5. Local thermodynamic equilibrium (LTE).
6. Volumetric power input due to ohmic heating.
7. Radiation heat losses can be treated as a volumetric heat sink.
8. The plasma is optically thin.
9. Negligible displacement currents.
10. No reactions occur inside or externally in the plasma reactor.

In the present work, the assumption of LTE is taken to be valid. Several works for micro-glow discharge plasmas [18–20] use a non-thermal equilibrium model where equations for electrons and heavy ions are solved. Further, a large number of reactions occurring in such processes are also solved. The present work deals with the following: 1. Atmospheric pressure thermal plasmas, 2. Torch dimensions as high as 250 mm (not micro-scale), 3. Low-velocity thermal plasmas. A good description of LTE v/s non-LTE has been provided by Punjabi et al. [21].

The boundary conditions are as follows:

- Inlet conditions ($z = 0$):

$$
\vartheta_z = \begin{cases}
Q_1/\pi \, r_1^2 & r < r_1 \\
0 & r_1 \le r \le r_2 \\
Q_2/\pi\left(r_3^2 - r_2^2\right) & r_2 \le r \le r_3 \\
0 & r_3 \le r \le r_4 \\
Q_3/\pi\left(R_0^2 - r_4^2\right) & r_4 \le r \le R_0
\end{cases} \tag{7}
$$

$$
\vartheta_r = 0 \tag{8}
$$

$$
\vartheta_\theta = \begin{cases}
0 & r < r_1 \\
0 & r_1 \le r \le r_2 \\
\vartheta_{\theta 2} & r_2 \le r \le r_3 \\
0 & r_3 \le r \le r_4 \\
\vartheta_{\theta 3} & r_4 \le r \le R_0
\end{cases} \tag{9}
$$

$$
T = 300 \, K \quad \frac{\partial A_R}{\partial z} = \frac{\partial A_I}{\partial z} = 0 \tag{10}
$$

$$
\kappa = 0.005(\vartheta_z^2 + \vartheta_\theta^2) \tag{11}
$$

$$
\varepsilon = 0.1\kappa^2 \tag{12}
$$

- Centreline ($r = 0$):

$$
\frac{\partial \, \vartheta_z}{\partial \, r} = \vartheta_r = \frac{\partial \, H}{\partial \, r} = \frac{\partial \, \kappa}{\partial \, r} = \frac{\partial \, \varepsilon}{\partial \, r} = A_R = A_I = 0 \tag{13}
$$

- Wall ($r = R_0$):

$$
\vartheta_z = \vartheta_r = k = \varepsilon = 0 \tag{14}
$$

$$
\lambda \frac{\partial \, T}{\partial \, r} = \frac{\lambda_w}{\delta_w}(T_s - T_w) \tag{15}
$$

$$
A_R = \frac{\mu_0 I}{2\pi}\sqrt{\frac{R_c}{R_0}}\sum_{i=1}^{coil} G(k'_i) + \frac{\mu_0 \omega}{2\pi}\sum_{p=1}^{C.V.}\sqrt{\frac{r_p}{R_0}}\sigma_p A_{I,p} S_p G(k'_p) \tag{16}
$$

$$A_I = -\frac{\mu_0 \omega}{2\pi} \sum_{p=1}^{C.V.} \sqrt{\frac{r_p}{R_0}} \sigma_p A_{R,p} S_p G(k'_p) \tag{17}$$

where,

$$G(k') = \frac{(2 - k'^2)K(k') - 2E(k')}{k'} \tag{18}$$

$$k'^2_p = \frac{4R_0 r_p}{(r_p + R_0)^2 + (z_b - z_p)^2} \quad k'^2_i = \frac{4R_i R_0}{(R_i + R_0)^2 + (z_i - z_b)^2} \tag{19}$$

- Exit,

$$\frac{\partial(\rho\, v_z)}{\partial z} = \frac{\partial v_r}{\partial z} = \frac{\partial H}{\partial z} = \frac{\partial A_R}{\partial z} = \frac{\partial A_I}{\partial z} = \frac{\partial \kappa}{\partial z} = \frac{\partial \varepsilon}{\partial z} = 0 \tag{20}$$

### 4.2. Equations for the Calculation of Impedance

In the present work, the flow, temperature, and electromagnetic equations are coupled with the properties inside the inductor and inside the plasma region. A user-defined function (UDF) was written to solve for impedance by Lesinski [22]. The formula used to calculate the resistance and the inductance of the torch are as follows:

$$R_T = \sum_{i=1}^{coil} \left( \frac{2\pi\, r_i}{\sigma_{coil} S_{coil}} + \frac{\omega^2 \mu_0{}^2}{2\pi} \times \sum_{p=1}^{C.V.} r_p \sigma_p A_{R,p}^x S_p \sqrt{\frac{r_p}{R_i}} G(k_{i,p}) \right) \tag{21}$$

$$L_T = \sum_{i=1}^{coil} \left( \sum_{n=1}^{coil} \Lambda_n - \frac{\omega \mu_0{}^2}{2\pi} \times \sum_{p=1}^{C.V.} r_p \sigma_p A_{I,p}^x S_p \sqrt{\frac{r_p}{R_i}} G(k_{i,p}) \right) \tag{22}$$

where

$$\Lambda_n = r_i \mu_0 \sqrt{\frac{r_n}{r_i}} G(k_{i,n}) \text{ if } i \neq n \tag{23}$$

$$= N^2 r_i P_0 \times 1 \times 10^{-9} \text{ if } i = n \tag{24}$$

where

$$P_0 = 4\pi \left\{ 0.5 \left[ 1 + \frac{1}{6}\left(\frac{D_c}{2r_i}\right)^2 \right] - 0.84834 + 0.2041\left(\frac{D_c}{2r_i}\right)^2 \right\} \tag{25}$$

$P_0$ is the shape correction factor [5], $r_i$ is coil radius, $D_c$ is the cross-sectional diameter of the coil wire, and $N$ is the number of coil-turns. $\sigma_{coil}$ is the electrical conductivity of the copper coil.

### 4.3. Method of Solution

The convergence of the two systems operating points, i.e., the ICP model (as explained in Punjabi et al. [18]) and the model for solving impedance, was obtained using the Semi-Implicit Method for Pressure Linked Equations (SIMPLE) algorithm. For calculation of the impedance, first, an initial guess of $R_T$ and $L_T$ was assumed. The power dissipation $P_{rms}$ in the plasma and frequency of the oscillation $f$ were then computed, and the new plasma impedance was recalculated using $P_{rms}$ as the input to the ICP model. From the solution of the ICP, new plasma resistance and inductance were calculated in Equations (9) and (10). These were again used in solving Equation (1). This procedure was repeated until it reached a constant value. A comprehensive grid sensitivity has already been covered in our earlier work, Punjabi et al. [21].

## 5. Results and Discussion

### 5.1. Comparison of the Present CFD Model with Literature

The CFD model predictions of the present work have been first compared with the literature data. For this, the same geometry and conditions of [10,11] have been considered. The experimental results of Merkhouf and Boulos [11] have been compared with the model predictions of the present work, see Figure 3A. The model predictions have then been compared with the model predictions of Kim et al. [10], see Figure 3B. The model predictions show a deviation of 4–5% (over prediction) compared to the experimental data of Merkhouf and Boulos [11] while the deviation is 10–15% (under prediction) compared to the model predictions of Kim et al. [10]. The under predictions of the present model are pronounced for higher powers in both the frequencies (2.4 MHz and 4.9 MHz). The predictions shown by the model of Kim et al. [10] are different from the predictions of the present model due to the assumptions by Kim et al. [10]. Most importantly, a laminar model has been used while in reality the flow is turbulent and demands the use of turbulent models.

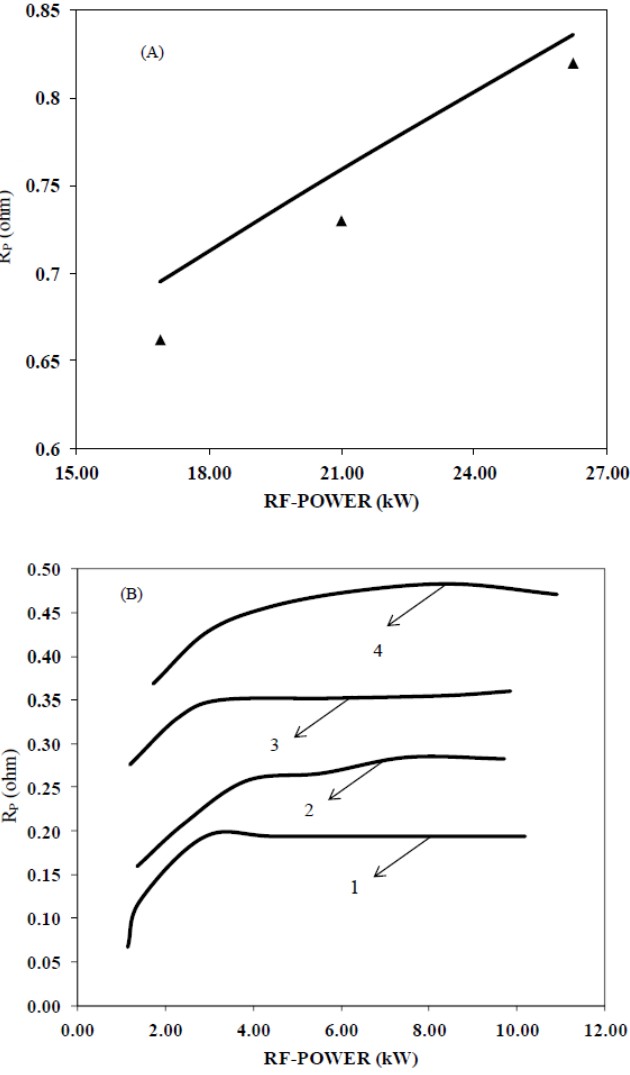

**Figure 3.** (**A**) Comparison of plasma resistance values obtained from Computational Fluid Dynamics (CFD) simulations using the present model with the experimental data from literature.—Present model ▲ Experimental data of [11]. (**B**) Comparison of plasma resistance values obtained from CFD simulations using the present model with the model of [10]. Present model: 1. 2.4 MHz; 3. 4.9 MHz Model in [10]; 2. 2.4 MHz; 4. 4.9 MHz.

*5.2. Comparison of Present CFD Predictions with Experimental Data*

5.2.1. Temperature Profiles

The present model predictions have been validated with in-house experimental data [21]. The radial temperature profiles at an axial location of 192 mm (in the centerline of the coil region) are shown in Figure 4A while axial temperature profiles are shown in Figure 4B. The temperature profile of the ICP is as shown in Figure 4 and it is evident that the temperature near the axis is higher than 5000 K. Further, the high temperature is up to 10 mm from the axis. There is a cold boundary layer at the wall, and the temperature at the quartz wall facing the plasma is less than about 800 K. Two sheath gas flowrates (10 lpm and 25 lpm) were used, and the plasma gas flowrate was maintained at 3 lpm. The power was 7.5 kW. A good agreement between CFD predictions and experimental data has been observed with a deviation of ±8%. Similarly, for a higher sheath gas flow rate of 25 lpm and power of 10 kW, as shown in Figure 4B, the deviation was found to be around 10%.

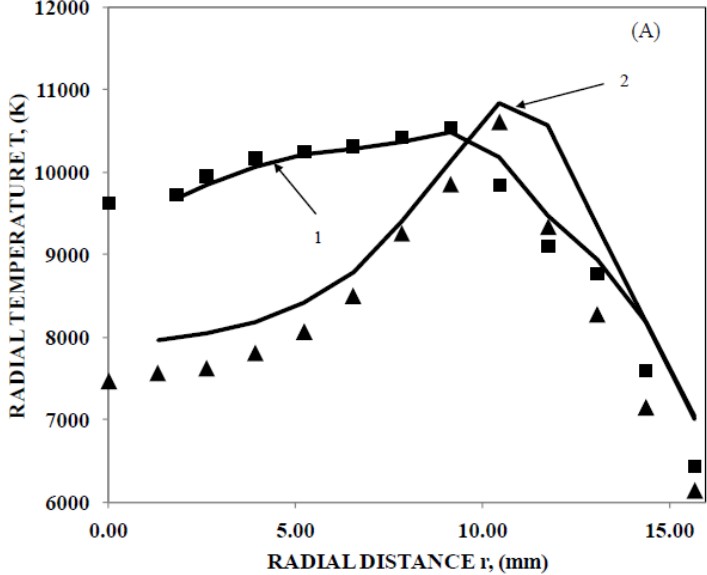

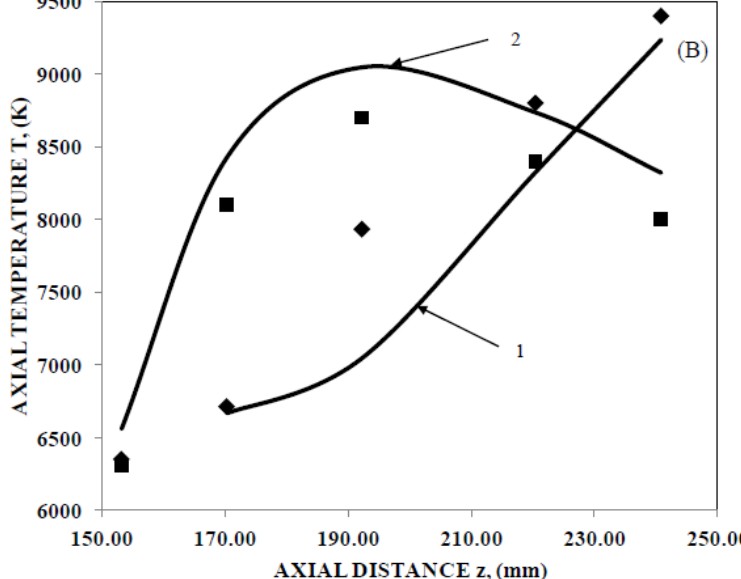

**Figure 4.** *Cont.*

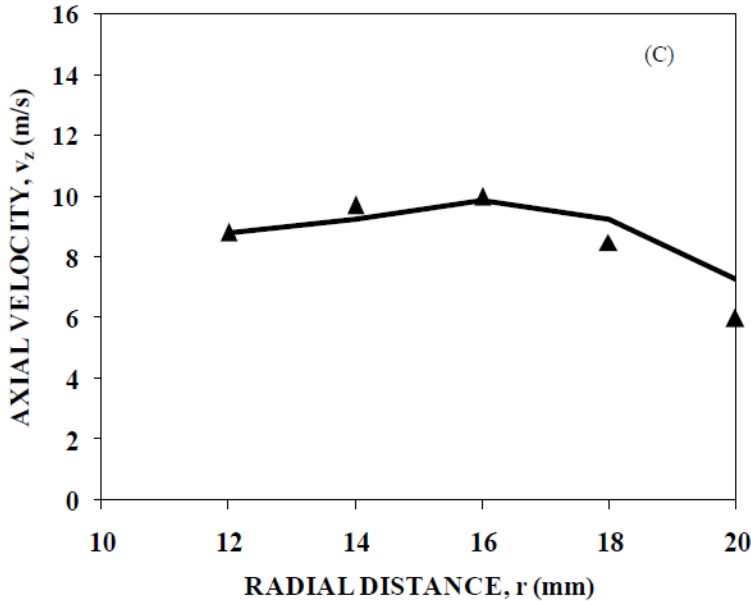

**Figure 4.** Temperature and velocity profiles: (**A**) Experimental and model-predicted axial temperature profiles for a plasma gas flow rate of 3 lpm and sheath gas flow rates of 10 and 25 lpm. ■ 10 lpm; ♦ 25 lpm; 1. 10 lpm; 2. 25 lpm. (**B**) Radial temperature profiles for a plasma gas flow rate of 3 lpm and sheath gas flow rates of 10 and 25 lpm. ■ 10 lpm; ♦ 25 lpm; 1. 10 lpm; 2. 25 lpm. (**C**) Velocity profiles for the comparison of experimental data of [22] with present model predictions for axial location z = 58 mm ▲ Experimental data—CFD predictions.

5.2.2. Velocity Profiles

The axial velocity profiles of the present model predictions were matched with the experimental data of Punjabi et al. [21] for the centerline (axial position of z = 58 mm as in Punjabi et al. [21]. The boundary conditions and geometry chosen were identical to the ones considered by Punjabi et al. [21]. A good agreement of 7–8% deviation has been observed between the model predictions and experimental results as shown in Figure 4C.

*5.3. Distribution of DC Power in Various Elements of RF Oscillator Circuit*

It was intended to study the effect of variation of power and sheath gas flow rate on the impedance of plasma. Therefore, three sets of measurements of $V_c$, $I_c$, and heat losses in various elements of the ICP system were taken with plasma gas flow rates of 3, 5, and 8 lpm. The sheath gas flow rate was varied as 10, 15, 21, 25, and 30 lpm, for each flow rate of plasma gas. To observe the effect of electrical power, the DC current fed to the oscillator was varied as 2, 3, 4, and 5 Amperes (A) for each combination of plasma gas and sheath gas flow rate. Calculation of resistance and inductance of the torch is performed as follows:

$$|Z_T| = \frac{V_c}{I_c} = |R_T + i\, X_T| \tag{26}$$

$$where \; R_T = |Z_T|cos\theta \quad X_T = |Z_T|sin\theta \quad L_T = \frac{X_T}{\omega} \tag{27}$$

Distribution of dissipated power in different elements of the RF power oscillator for various DC powers is as shown in Figure 5. It shows that as the DC power increases, the percentage of RF power produced by the oscillator increases and the percentage of power lost in the plate and the grid decreases.

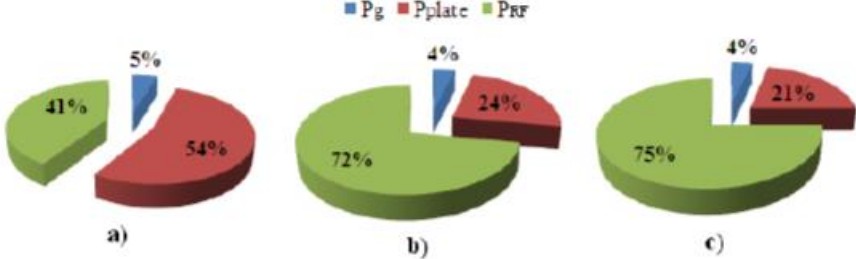

**Figure 5.** Distribution of dissipated powers in various elements of the RF power oscillator at (**a**) 3.75 kW (**b**) 12.25 kW and (**c**) 21.25 kW DC powers.

For the CFD part, the numerical code was run for approximately the same RF power, plasma flow rate, and sheath gas flow rate as that of the experimental run.

*5.4. Variation of Oscillator Efficiency with DC Power*

Figure 6 shows the variation of the oscillator efficiency as a function of the DC power supplied. A typical trend in the variation of the oscillator efficiency with DC power suggests that, as the power increases, the oscillator efficiency increases to a certain value and then remains more or less constant. The reason is as follows: For an RF system, a certain amount of DC power is required in maintaining the oscillator tube at the operating point and the rest is available as an RF output. Hence, when DC power is low, a very small amount of power is available as the RF output and the efficiency is low at low powers. As the applied DC power increases, the RF power produced by the oscillator also increases. As the DC power is increased beyond a certain level, the operating point of the oscillator shifts, the ambient plate current increases and the gain of the tube saturates. After this, the efficiency of the oscillator tends to saturate with DC power.

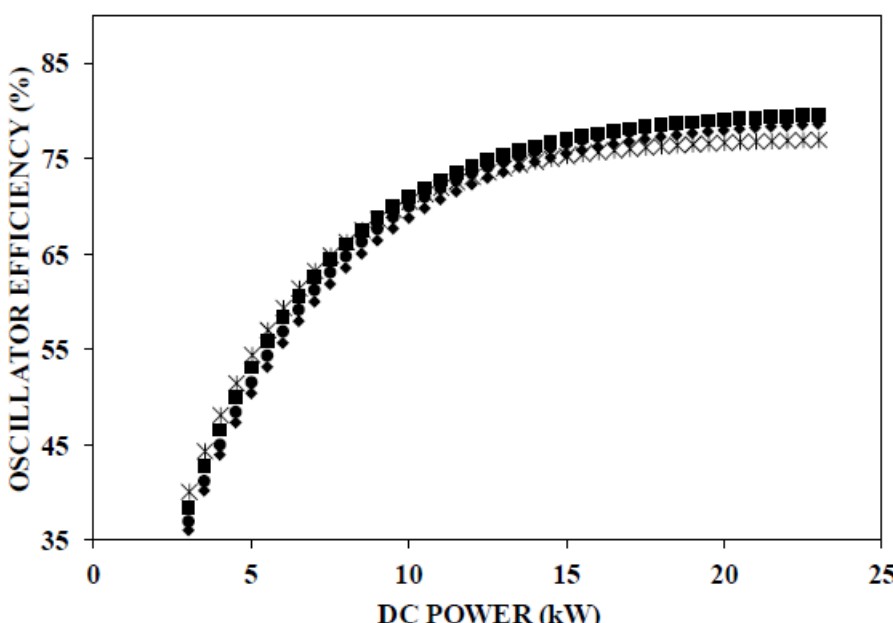

**Figure 6.** Experimental measurements of oscillator efficiency as a function of DC power with plasma gas at 8 lpm and sheath gas flow rate varied from 10 lpm to 30 lpm. ■ 15 lpm ♦ 21 lpm ● 25 lpm ✳ 30 lpm.

Figure 6 shows the variation of the efficiency with DC input power for different combinations of plasma and sheath gas flow rates. The results suggest that the power losses due to grid and plate have an important role to play when the DC power is varied.

### 5.5. Variation of Plasma Resistance and Inductance

In this section, we present the effect of sheath gas flowrate and RF power on the shape and volume of plasma, as well as on plasma resistance and torch inductance. Figure 7 presents the photograph of the behavior of plasma for different power inputs (for constant plasma and sheath gas flows). Figures 8–11 present the experimental and model results of plasma resistance and torch inductance for varying power inputs and sheath gas flow rates. In this section, the experimental results are compared with the model predictions. Deviations of ~20% of the model predictions are observed from the experimental data. This is attributed to the error (15%) associated with the measurement of the quantities as explained in Section 3.3. The effective deviations are observed to be about 5%.

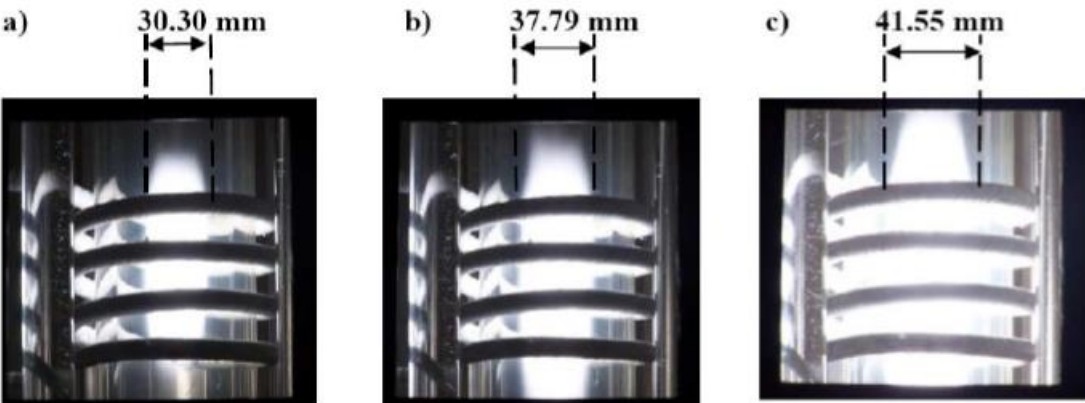

**Figure 7.** Photographs of plasma for (**a**) 6 kW (**b**) 9.75 kW, and (**c**) 20.25 kW DC power. Plasma gas and sheath gas flows were held constant at 5 and 30 lpm, respectively.

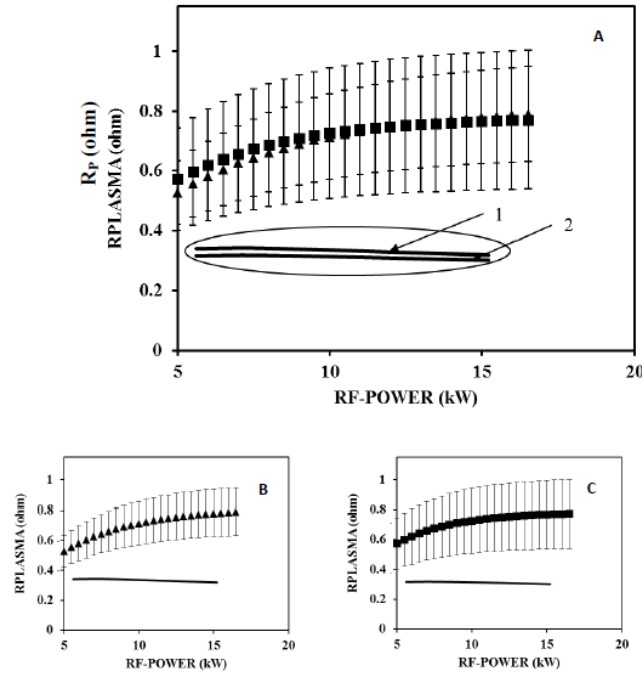

**Figure 8.** (**A**) Plasma resistance as a function of RF power with plasma gas at 5 lpm and a sheath gas flow rate of 10 lpm and 30 lpm. Experimental data ▲10 lpm ■ 30 lpm. Model-predicted results: 1. 10 lpm; 2. 30 lpm. (**B**) Variation of plasma resistance vs. RF power for plasma gas flow of 5 lpm and sheath gas flow of 10 lpm from both experimental data (clearly showing error bars) and predicted results. (**C**) Variation of plasma resistance vs. RF power for a plasma gas flow of 5 lpm and sheath gas flow of 30 lpm from both experimental data (clearly showing error bars) and predicted results.

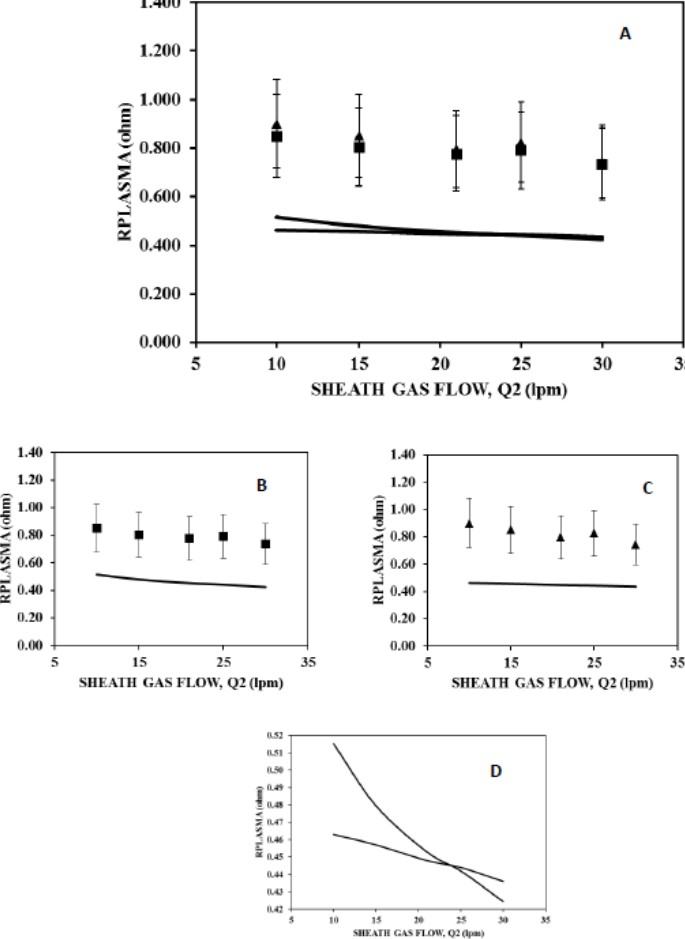

**Figure 9.** (A) Plasma resistance as a function of sheath gas flow rate for a plasma gas flow rate of 3 lpm at powers of 10 kW and 15 kW. Experimental data ■ 10 kW ▲ 15 kW Model Predicted results: 1. 10 kW; 2. 15 kW. (**B**) Variation of plasma resistance vs. sheath gas flow for the plasma gas flow of 3 lpm and RF power of 10 kW from both experimental data (clearly showing error bars) and predicted results. (**C**) Variation of plasma resistance vs. sheath gas flow for a plasma gas flow of 3 lpm and RF power of 15 kW from both experimental data (clearly showing error bars) and predicted results (**D**) Comparison of only model predicted results for both the above cases. Zoomed to show appropriate variation

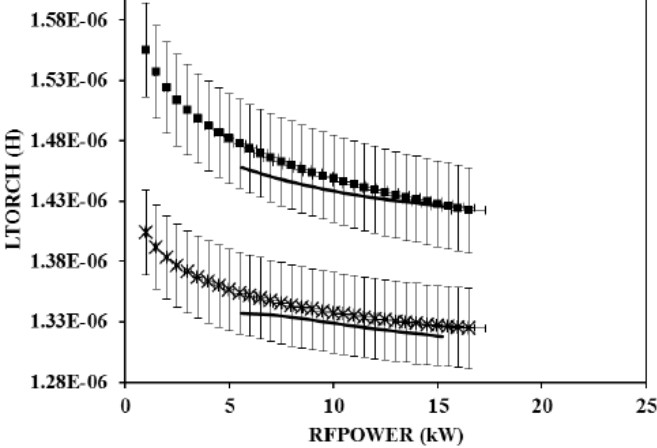

**Figure 10.** Inductance of a torch as a function of RF power with plasma gas at 5 lpm and a sheath gas flow rate of 10 lpm and 30 lpm. ■ 10 lpm; ✳ 30 lpm; 1. 10 lpm; 2. 30 lpm.

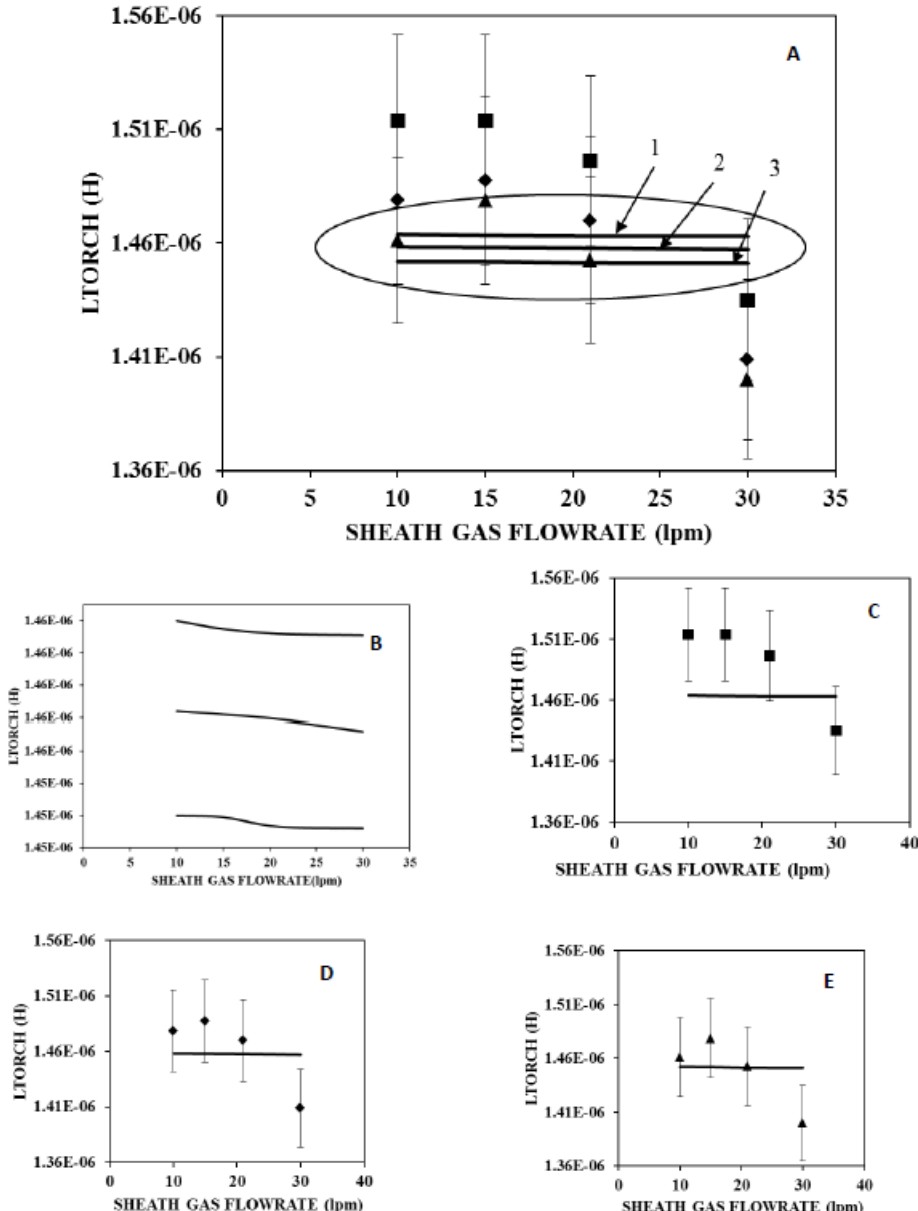

**Figure 11.** (**A**) Torch inductance as a function of sheath gas flow at RF powers 5, 10, and 15 kW with a plasma gas flow of 3 lpm. Experimental data: ■ 5 kW; ♦ 10 kW; ▲ 15 kW. Model results: 1. 5 kW; 2. 10 kW; 3. 15 kW. (**B**) Zoomed portion of the model-predicted results (**C**) Variation of torch inductance vs. sheath gas flow at RF power of 5 kW from both experimental data (clearly showing error bars) and predicted results. (**D**) Variation of torch inductance vs. sheath gas flow at RF power of 10 kW from both experimental data (clearly showing error bars) and predicted results. (**E**) Variation of torch inductance vs. sheath gas flow at RF power of 15 kW from both experimental data (clearly showing error bars) and predicted results.

### 5.5.1. Variation of Resistance

Plasma resistance depends on two factors viz. plasma volume and plasma temperature, both of which depend on applied RF power and sheath gas flow rate. In the preceding subsections, we present a parametric study of the variation of resistance.

Effect of Power

The temperature, shape, and volume of plasma change with gas flow rate and RF power. In general, increasing the RF power increases the plasma temperature, diameter, and length. At low powers, the plasma diameter is small compared to that of the quartz tube containing plasma. Therefore, by increasing the RF power, the plasma diameter and thus volume can freely increase, which increases the plasma resistance, as seen in Figure 8. The plasma temperature also increases to some extent which decreases plasma resistance. The net effect is an increase in the plasma resistance because of the effect of the increase in plasma volume exceeding that of the increase in plasma temperature.

At high-power inputs, the plasma diameter is large and increasing the power does not lead to a significant increase in the plasma diameter. This is due to the finite size of the quartz tube containing plasma. However, the temperature increases leading to a reduction in resistance. Therefore, the effect of increase in plasma volume is balanced by the effect of increase in temperature.

CFD simulations using the turbulent flow model were performed with a plasma gas flow of 3 lpm and sheath gas flows of 10 lpm and 30 lpm. The inlet swirl velocities for 10 lpm and 30 lpm sheath gas flow rates were 23.58 m/s and 70.77 m/s, respectively. The effect of RF power on the plasma resistance and torch inductance was investigated, as shown in Figure 8. It can be clearly observed that both CFD simulations and experimental data represent identical trends. In both the experimental and CFD results, the plasma resistance tends to saturate beyond an RF power of 10 kW. The discrepancy in the values of plasma resistance obtained from CFD and experimental results is 10–15%.

Effect of Sheath Gas Flow Rate

The effect of sheath gas flow on plasma resistance is slightly different from that of the RF power. An increase in the sheath gas flow changes the temperature distribution in the plasma. This leads to a change in the shape of plasma. Particularly, plasma extends upstream, and one can see the luminous zone extending well above the first turn of the coil, as in our earlier work [18]. An increase in the plasma temperature leads to an increase in its electrical conductivity and its inductance. It was observed that the plasma volume decreases with an increase in the sheath gas flow, but the plasma diameter does not change significantly. The physical significance can be explained as follows: As the sheath gas flow increases, two things happen: (1) plasma volume (plasma length) decreases, hence the plasma resistance decreases. (2) The axial temperature at upstream points (above the first turn of the coil) increases.

In addition to the photographs, we also present the profiles of plasma resistance. Figure 9A shows the profiles of plasma resistance with an increase in the sheath gas flow rate (a zoomed view and comparison of experimental and predicted results for each RF power is shown in Figure 9B–D). The combined effect of plasma volume and temperature leads to a decrease in the plasma resistance with an increase in the sheath gas flow. The variation of the plasma resistance with power and sheath gas flow at plasma gas flow rates of 3, 5, and 8 lpm is similar.

5.5.2. Variation of Torch Inductance

Plasma inductance depends on how much magnetic flux produced by the coil is linked with plasma. Flux linkage depends on electrical conductivity (which depends on temperature) and the volume of plasma. Thus, plasma inductance depends on the gas flow rate and RF power. Torch inductance is related to plasma inductance, as in Equation (1). We now proceed to discuss the effect of power and gas flow rates on the inductance of plasma.

Effect of RF Power

Figure 10 shows that both in-house experimental data and predicted-model results confirm that torch inductance decreases with an increase in RF power. Further, as plasma volume (diameter)

increases with RF power, the flux leakage between the plasma and the coil decreases and $L_P$ increases. Thus, the resultant torch inductance ($L_{coil} - L_P$) decreases.

Effect of Sheath gas Flow Rate

Since plasma inductance depends strongly on its conductivity, torch inductance decreases with an increase in the sheath gas flow, as shown in Figure 11. Figure 11 illustrates the variation of torch inductance with the sheath gas flow rate. A zoomed view and comparison of the experimental and predicted results for each RF power is shown in Figure 11B–E. It shows that as the sheath gas flow rate increases, the temperature at the upstream of the coil location increases, as shown in Figure 11, and, therefore, electrical conductivity also increases. This results in a decrease of the torch inductance with an increase in the sheath gas flow. A similar trend of the inductance is observed for a plasma gas flow rate of 5 and 8 lpm.

## 6. Conclusions

The variation of plasma resistance, inductance, and efficiency of an ICP torch with different power and flow rates was studied. For the ICP system, the oscillator efficiency ranges from 40% to 80% for DC power in the range of 3.75 kW to 24 kW. Plasma resistance depends on the plasma volume and plasma temperature. It is seen that plasma resistance increases as a function of power and decreases as a function of the sheath gas flow. Torch inductance decreases as a function of power. This is because as the power increases, the plasma volume increases and the mutual inductance between plasma and coil also increase ($L_P$ increases). Torch inductance decreases with an increase in the sheath gas flow. This is because, as the sheath gas flow increases, the plasma temperature increases from upstream locations and the average temperature goes up. This increases the electrical conductivity of the plasma which, in turn, decreases the torch inductance. This paper shows that the plasma impedance and efficiency of the system depend on the amount of power applied, plasma gas flow rate, and sheath gas flow rate. A numerical model for ICP can be validated by these experimental results and can be used for scaling up the power and for designing a new ICP system.

**Author Contributions:** The authors D.N.B., N.K.J., A.K.D., D.C.K. provided able guidance for the design and operation of experiments. The authors S.B.P. and S.N.S. have conceptualized, designed the experimental set-up and performed experiments. A.A.G. with S.P. has carried out the modeling part by extensively performing simulations for literature and experimental points. Writing the paper with guidance of J.B.J. was majorly done by A.A.G. J.B.J. has provided his overall guidance of conceptualization to completion of the paper and has played a pivotal role in the modeling work.

**Funding:** The research has been funded by BRNS which is the funding body for Nuclear Scientific Research of the Government of India.

**Acknowledgments:** The authors are thankful to L.M. Gantayet, Director, Beam Technology Development Group, for his support during the course of this work. This work was made possible through continuing research grants from BRNS. We are thankful to D.S. Patil for making the system operational at high powers and making it available for us. We are also thankful to P.K Soni, D.P. Chopade, S.T. Thakur and D.K. Baskey for their technical support. The fellowship given by BRNS to Sangeeta Punjabi during the course of this work is gratefully acknowledged.

**Conflicts of Interest:** The authors declare that there is no conflict of interest.

## Nomenclature

| | |
|---|---|
| $A_I$ | Imaginary part of vector potential in Equation (19) |
| $A_R$ | Real part of vector potential in Equation (18) |
| $A_{R,p}$ | Real part of vector potential for pth volume as in Equation (19) |
| $A_{I,p}$ | Real part of vector potential for pth volume as in Equation (18) |
| $A_\theta$ | Vector potential in azimuthal direction |
| $A_{ul}$ | Transition probability between the upper level $u$ and the lower level $l$ |

| | |
|---|---|
| B | Constant |
| c | velocity of light as in Equation (3) |
| $C_\mu$ | turbulence model constant |
| $C_1$ | turbulence model constant |
| $C_2$ | turbulence model constant |
| C1 | high-voltage capacitor which are part of RF oscillator (nF) |
| C2 | high-voltage capacitor which are part of RF oscillator (nF) |
| $C_{\varepsilon_1}$ | constant used in turbulent energy dissipation equation Table 4, (-) |
| $C_{\varepsilon_2}$ | constant used in turbulent energy dissipation equation Table 4, (-) |
| $C_p$ | Heat capacity of water (kJ kg$^{-1}$ °C$^{-1}$) |
| dc | coil tube diameter |
| e(t) | induced voltage at the terminal of Rogowski coil (V) |
| $E_u$ | excitation energy of the upper level u |
| $E_\theta$ | electric field in azimuthal direction |
| $E(k')$ | complete elliptic integrals |
| $f$ | resonating frequency (MHz) |
| $F_r$ | radial body force |
| $F_z$ | axial body force |
| $G_\kappa$ | production of turbulent kinetic energy |
| $G(k')$ | function of complete elliptical integrals |
| $G(k_p)$ | function of complete elliptical integrals at pth volume |
| | enthalpy |
| $H_r$ | radial component of magnetic field |
| $H_z$ | axial component of magnetic field |
| $H_z^\times$ | complex conjugate of axial component of magnetic field |
| I | Coil current |
| $I_c$ | RMS RF current across the coil |
| $I_{dc}$ | DC current |
| $I_g$ | DC grid current |
| $I_{Rog}$ | current across Rogowski coil |
| i | imaginary as in Table 4, (i = $\sqrt{-1}$) |
| K | Complete elliptic integrals |
| $K(k')$ | Complete elliptic integrals |
| k | Boltzmann constant as used in Equation (2) |
| κ | Turbulent kinetic energy as in Table 4 |
| $k'$ | constant depending on radial and axial co-ordinates Equations (18)–(21) |
| $k'_i$ | constant as in Equation (21) for ith volume |
| $k'_p$ | constant as in Equation (21) for pth volume |
| $L_C$ | Inductance of the coil |
| $L_g$ | Grid choke |
| $L_P$ | Inductance of the plasma |
| $L_T$ | Inductance of the torch |
| $\dot{m}_i$ | mass flow rate |
| N | number of coil turns |
| P | Input power |
| $p$ | pressure |
| $P_{dc}$ | DC power |
| $P_g$ | Power loss in grid circuit |
| $P_i$ | Power loss in cooling water through each element i |
| $P_{plate}$ | Power loss in plate circuit of the triode |
| Q1 | central injection gas flow rate |
| Q2 | plasma gas flow rate |
| Q3 | sheath gas flow rate |
| $R_C$ | Resistance of the coil |
| $R_g$ | Grid leak resistance |

| | |
|---|---|
| $R_P$ | Reflected resistance of plasma as seen by the coil |
| $R_{sh}$ | transducer gain |
| $R_T$ | Resistance of the torch |
| R0 | Radius of the confinement tube |
| Rc | Radius of the coil |
| $R_i$ | Radius of ith coil |
| S | cross-sectional area of the coil |
| Sp | Cross-section of the pth control volume |
| S | averaged strain term used in production term as in Table 4 |
| $S_{ij}$ | Strain term as in Table 4 |
| $|S|$ | Absolute value of averaged strain term used in production term as in Table 4 |
| $\left|S_{ij}\right|$ | Absolute value of strain term as in Table 4 |
| $T$ | Temperature |
| $T_s$ | Inside surface temperature of quartz tube |
| $T_{in}$ | Inlet temperature of cooling water |
| $\Delta T_i$ | Temperature rise across each element i |
| Ti | integrator time constant |
| $T_o$ | Outlet temperature of cooling water |
| $U_p$ | local energy dissipation rate |
| $U_R$ | Volumetric radiation heat losses |
| $\vartheta_r$ | Radial component of velocity |
| $\vartheta_\theta$ | Tangential (swirl) component of velocity in Equation (11) |
| $\vartheta_{\theta 2}$ | Tangential (swirl) component of velocity for a particular radius range as in Equation (11) |
| $\vartheta_{\theta 3}$ | Tangential (swirl) component of velocity for a particular radius range as in Equation (11) |
| $\vartheta_z$ | Axial component of velocity |
| $V_{dc}$ | DC voltage |
| $V_c$ | RMS RF voltage across the coil |
| Vm | output voltage at the terminal of the current transducer |
| $X_T$ | Reactive part of the torch |
| $Z_T$ | Impedance of inductively coupled plasma torch |
| $z$ | Distance in axial direction |
| $z_b$ | Height of the boundary at pth control volume. |
| $z_i$ | Height of the ith coil |
| **Greek symbol** | |
| $\alpha_{eff}$ | Thermal diffusivity |
| $\delta_w$ | Tube wall thickness |
| $\lambda$ | Thermal conductivity |
| $\lambda_w$ | Thermal conductivity of the quartz confinement tube ($\lambda_w$ = 1.047 W/mK) |
| $\mu_0$ | Permeability of free space |
| $\omega$ | Angular frequency |
| $\rho$ | Density |
| $\mu$ | kinematic viscosity |
| $\nu$ | dynamic viscosity |
| $\sigma$ | Electrical conductivity |
| $\sigma_p$ | Electrical conductivity at pth control volume |
| **Subscript** | |
| C | coil |
| dc | direct current |
| g | grid |
| i | element |
| in | inlet |
| o | outlet |
| P | Plasma |
| T | Torch |



**Abbreviation**

| | |
|---|---|
| 2D | Two dimensional |
| DSO | Digital Storage Oscilloscope |
| ICP | Inductively Coupled Plasma |
| ID | Inner diameter |
| OD | Outer diameter |
| RF | Radio frequency |

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
