# Peer review of "Computational Fluid Dynamics (CFD) Simulations and Experimental Measurements in an Inductively-Coupled Plasma Generator Operating at Atmospheric Pressure: Performance Analysis and Parametric Study"

_processes, doi:10.3390/pr7030133_

Round 1

Reviewer 1 Report

Summary:  This manuscript reports CFD modeling (turbulent modeling) of an atmospheric pressure plasma torch with argon gas.  A comparison between various operating parameters (including gas flowrate, and RF/DC power) and plasma properties are shown.

Major Concerns/Comments:

It is unclear which equations are being solved.  If a paper describes CFD studies, the basic governing equations being solved must be included somewhere in the paper.  A number of the parameters listed in the nomenclature never show up in any equations, so it's misleading to have all of them in the paper.  Additionally, if a turbulent model is used, what are the turbulent model parameters/conditions?

It would be helpful to the reader to understand how the plasma modeling is done.  For example, were any reactions accounted for (plasma chemistry), was an external circuit included in the modeling, etc.?  If non of these aspects were included, how is the plasma being modeled?  See work by Farouk, T. I., et al. (PSST 2007 and 2008, J. Phys. D 2008 and IEEE Trans. Plasma Sci. 2010)

The figures need a lot of improvement.  It is unclear what is being compared at times as the captions appear incorrect/misleading/difficult to read.  See caption in Figure 3, among others.

The temperatures discussed here are very high >5000 K.  How are these temperatures being measured?  And how is the setup operating at such high temperatures?  Quartz (based on this reviewer's knowledge) melts at 1500-1700 oC, how is this setup functioning at such high temperatures? 

Water is said to be flowed between the concentric quartz cells that make up the plasma torch cell.  What is the effect of water flow on the plasma?  Water is not perfectly dielectric and water flow can generate its own inductive fields in the presence of an RF electric field.  Also, how is this water not boiling at such high temperatures?

In the abstract, "very good agreement" is claimed between the simulations and experimental measurements (literature and current work), however, the comparisons between experiments and simulations in Figures 8-11 are not very good.  Can this be clarified?

The uncertainty/error analysis shown for the power seems incorrect.  Can it be clarified where this was obtained from?

Additional Comments:

Page 1, line 41: The sentence does not make sense.  Please clarify what "unknowns" are being referred to.

Table 1: There are a number of "numbers" in the last four columns, however it is difficult to know what they refer to.  Perhaps a note can be added to the table referring the reader to the end of the manuscript where these numbers are explained.

Figure 1 is placed after figure 2.  Is there any reason for this?  Figures are typically listed in the order they are referenced in the text.

Why was an incompressible flow model used for the plasma?  The plasma gas can be highly compressible.  A justification for such an assumption must be included.

Author Response

We thank the reviewer for the valuable comments and suggestions. In view of these, we have revised the manuscript. The rebuttals to the comments and the details pertaining to the changes made in the revised manuscript have been attached in the word document

Reviewer 2 Report

Title "Computational Fluid Dynamics" is better for someone who does not understand CFD.

In Figure 1 , presentation to the scale is requested: Outer diameter 80 mm and water chanell innner tube outer diameter 60 mm. Length is 500 mm. beginning of the plasm is 150 mm and the end is 250 mm from the Top flange. Please show as it is.

Layout of Fig. 11 can be better improved.

In Figure 8 and 9, if the discrepance is the expansion. It can be adjusted empirically and add to the discussion.

The error bar is overlapped.

Introduction seems to be too long. Is overall review in the present form necessary.

There is two conclusions: in page 18 line 315 and in page 20 line 437 to page 21 line 446.

Assumption and FIndings Limitation in page 20 are queer. 

Author Response

We thank the reviewer for the valuable comments and suggestions. In view of these, we have revised the manuscript. The rebuttals to the comments and the details pertaining to the changes made in the revised manuscript can be found in the word document.
